# Immunohistochemical Analysis of Spermatogenesis in Patients with SARS-CoV-2 Invasion in Different Age Groups

Grigory A. Demyashkin [1,2,*] , Evgeniya Kogan [2], Tatiana Demura [2], Dmitry Boldyrev [2] , Matvey Vadyukhin [2] , Vladimir Schekin [1,2], Peter Shegay [1] and Andrey Kaprin [1]

[1]   National Medical Research Radiological Centre of the Ministry of Health of the Russian Federation, Obninsk 249036, Russia
[2]   Department of Clinical Anatomy and Operative Surgery, Department of Pathological Anatomy, Institute for Urology and Reproductive Health, Sechenov University, Moscow 119991, Russia
*   Correspondence: dr.dga@mail.ru

**Abstract:** Based on studies that focused on the effect of SARS-CoV-2 on human tissues, not only pulmonary invasion was revealed, but also impaired testicular function. Thus, the study of the mechanisms of influence of SARS-CoV-2 on spermatogenesis is still relevant. Of particular interest is the study of pathomorphological changes in men of different age groups. The purpose of this study was to evaluate immunohistochemical changes in spermatogenesis during SARS-CoV-2 invasion in different age groups. In our study, for the first time, a cohort of COVID-19-positive patients of different age groups was collected, and the following were conducted——confocal microscopy of the testicles and immunohistochemical evaluation of spermatogenesis disorders in SARS-CoV-2 invasion with antibodies to the spike protein, the nucleocapsid protein of the SARS-CoV-2 virus, and angiotensin convertase type 2. An IHC study and confocal microscopy of testicular autopsies from COVID-19-positive patients revealed an increase in the number of S-protein- and nucleocapsid-positively stained spermatogenic cells, which indicates SARS-CoV-2 invasion into them. A correlation was found between the number of ACE2-positive germ cells and the degree of hypospermatogenesis, and in the group of patients with confirmed coronavirus infection older than 45 years, the decrease in spermatogenic function was more pronounced than in the cohort of young people. Thus, our study found a decrease in both spermatogenic and endocrine (Leydig cells) testicular functions in patients with COVID-19 infection. In the elderly, these changes were significantly higher than in the group of young patients.

**Keywords:** testicles; viral orchitis; hypospermatogenesis; SARS-CoV-2; COVID-19; immunohistochemistry

## 1. Introduction

A pandemic of coronavirus infection caused by the SARS-CoV-2 virus was declared by the WHO in March 2020 [1]. Based on the accumulated knowledge and new methods for studying the mechanisms of the influence of this pathogen on human tissues, not only was pulmonary invasion was revealed, but also a violation of the functions of other organs and systems, including the testicles [2].

Researchers found signs of viral orchitis in testicles with a predominant lesion of germ cells by apoptosis, inflammatory infiltration of interstitial tissue, and basement membrane sclerosis in patients who had SARS-CoV infection in 2002 [3]. Similar results were obtained with a new COVID-19 infection with similar structural and functional disorders in the testicles: an increase in the number of apoptotic cells, CD138+ plasmacytes, CD3+ T-lymphocytes, CD20+ B-lymphocytes, and CD68+ macrophages in the interstitium [4].

According to the results of single studies, it was found that the level of receptors for angiotensin-converting enzyme 2 (ACE2) was significantly lower in the testicles of patients with a confirmed COVID-19 infection and with physiological spermatogenesis

when compared with a group of patients with pathological spermatogenesis [5]. Thus, it can be concluded that an increase in the level of ACE2 expression directly correlates with the degree of impaired spermatogenesis. However, most of the studies were conducted with a small sample of men, and the results were contradictory [4,6,7].

Thus, the study of the mechanisms of influence of SARS-CoV-2 on spermatogenesis is still relevant. Of particular interest is the study of pathomorphological changes in men of different age groups, information about which is not available [5,8].

It is known that the SARS-CoV-2 virus enters cells by binding to cellular ACE2 receptors through the receptor-binding domain in the S-protein, which is confirmed by immunohistochemical reactions with antibodies to the S-protein and nucleocapsid, as well as to ACE-2 [9,10].

Attempts have been made to study the expression of the ACE2 and TMPRSS2 genes, but the results have been controversial. Despite an increase in the expression of the ACE2 gene according to the results of a molecular study, an increase in the expression of ACE2 and TMPRSS2 proteins in spermatogonia was not found [11,12]. Thus, in a large cohort of patients, SARS-CoV-2 invasion into spermatogenic cells still remains to be clarified.

In our study, for the first time, a cohort of COVID-19-positive patients of different age groups was collected.

The aim of this study was to evaluate immunohistochemical changes in spermatogenesis during SARS-CoV-2 invasion in different age groups.

## 2. Material and Methods

### 2.1. Experiment Design

According to the WHO age periodization and anamnestic, clinical, and morphological data, the following groups were formed:

Group I (*n* = 90; age 25–90 years). The cause of death is a new coronavirus infection caused by the SARS-CoV-2 virus (PCR+), with the development of bilateral total pneumonia of mixed genesis. The disease was complicated by acute respiratory distress syndrome in adults (diffuse alveolar damage) and multiple organ failure. The disease was complicated by acute respiratory distress syndrome in adults (diffuse alveolar damage) and multiple organ failure. COVID-19 cases have been defined according to the World Health Organization (WHO) as cases with laboratory confirmation of SARS-CoV-2 infection, regardless of clinical signs and symptoms (Table 1).

**Table 1.** Distribution by groups, according to the age periodization of men (WHO, 2014).

| Age | Subgroup | *n* | Age | Subgroup | *n* |
|---|---|---|---|---|---|
| 25–44 (young) | I—COVID-19 | 9 | 61–75 (elderly) | I—COVID-19 | 43 |
| | II—control | 5 | | II—control | 5 |
| 45–60 (middle) | I—COVID-19 | 17 | 76–90 (senile) | I—COVID-19 | 21 |
| | II—control | 5 | | II—control | 5 |

Group II (*n* = 20). Normal testicular autopsies obtained no later than 6 h after biological death were declared. There were no macroscopic signs of an inflammatory and/or tumor process; each of the patients had at least one child. Patients in this group had no previous exposure to toxic substances. Causes of death in elderly and senile men—postinfarction cardiosclerosis, chronic cerebral ischemia; in the middle-age group—gastric ulcer with perforation and/or bleeding, obstructive pyelonephritis, pancreatitis, aneurysms of various localization; in young people—congenital malformations, but not of the genitourinary system (SARS-CoV-2, PCR–) (Table 1).

### 2.2. Ethics Approval

The study was conducted in accordance with ARRIVE guidelines and the requirements of the Declaration of Helsinki (1964), U.K. Animals (Scientific Procedures) Act (1986), Pri-

vacy Act (1988), and was approved by the Local Ethical Committee of Sechenov University (protocol No. 162 of 11 February 2022). Research was pursued with appropriate informed consent of participants or guardians and justified by its potential benefit, and conducted by qualified staff using appropriate methods and resources. Informed consent was obtained from relatives of all deceased patients who could not be identified from this study.

### 2.3. Exclusion Criteria

Infertility, endocrine diseases, obesity, bacterial infection, sepsis, HIV infection, viral hepatitis B and C, Epstein-Barr virus, mumps, atherosclerosis, arterial hypertension, diabetes mellitus, chronic alcoholism, and drug addiction.

### 2.4. Morphological Study

After extraction, the appearance of the testicles and the condition of the parenchyma on the section were evaluated, weighed (in grams), and measured. They were then cut parallel to the sagittal plane every 2 mm, fixed in formalin, and, after wiring (tissue histological wiring machine, Leica Biosystems, Wetzlar, Germany), embedded in paraffin blocks, from which serial sections were prepared (3 μm thick), deparaffinized, dehydrated, and stained with hematoxylin and eosin for histological examination.

### 2.5. Immunohistochemical Study

This was performed in automatic mode (BOND-III Fully Automated IHC and ISH Staining System, Leica, Wetzlar, Germany). Rabbit polyclonal antibodies to the spike protein (SARS-CoV-2 Spike Antibody, GeneTex, Irvine, CA, USA, 1:500), nucleocapsid protein (SARS-CoV-2 Nucleocapsid Antibody, GeneTex, CA, USA, 1:500) of the SARS-CoV-2 (COVID-19), and angiotensin convertase type 2 (ACE2, GeneTex, CA, USA, 1:250) were used as primary antibodies.

The number of immunopositive cells was counted in 10 randomly selected fields of view at $\times 400$ magnification (in percent). Microscopic analysis was performed using a video microscopy system (microscope Leica DM2000, Wetzlar, Germany; camera Leica ICC50 HD).

### 2.6. Fluorescence In Situ Hybridization

Similar to IHC, probe detection has been achieved using a fluorescent technique called fluorescence in situ hybridization (FISH). To study the samples using confocal microscopy, testis fragments were fixed with 2% formaldehyde in PBS for 1 h, soaked in 30% sucrose solution (which served as a cryoprotectant), and frozen in isooctane cooled with liquid nitrogen. The samples were kept at $-30$ °C. Sections 7–10 μm thick were obtained using a Leica cryostat (Germany). The studied markers were detected by indirect immunofluorescence. Sections were permeabilized with 0.3% Triton X-100 in PBS for 20 min. Nonspecific binding of immunoglobulins was blocked with 3% bovine serum albumin (BSA) in buffer solution for 1 h. Sections were incubated with primary antibodies overnight at 4 °C, after which the sections were washed 4 times for 10 min in a buffer solution with 0.5% BSA. Anti-mouse immunoglobulins conjugated with fluorescent dyes were used as secondary antibodies: Alexa Fluor 488 (Thermo Fisher Scientific, Waltham, MA, USA) at a dilution of 1:500 buffer solution with 0.5% BSA, with which sections were stained for 2 h at room temperature. After that, the sections were washed 4 times for 30 min in a buffer solution with 0.5% BSA. Sections embedded in glycerol were analyzed using a Zeiss LSM 5 Pascal confocal microscope with argon and helium-neon lasers (Carl Zeiss, Oberkochen, Germany).

### 2.7. Statistical Analysis

The obtained data were processed using the SPSS 12.0 software (IBM Analytics, Armonk, NY, USA). All data are presented as M $\pm$ m. The hypothesis of normal distribution of values in the samples was tested using the Kolmogorov–Smirnov test, after which the

Student's *t*-test for small samples, the nonparametric Mann–Whitney *U*-test, and Fisher's exact test were used. Differences between samples were considered statistically significant at $p < 0.05$.

## 3. Results

Macroscopic examination and testicular weight in patients with COVID-19 did not differ from those in the control group. Testicular autopsies were performed to rule out obstruction of the seminal ducts and to identify the cause and extent of spermatogenesis disorders.

Histological examination. Microscopic analysis of the testicles of patients with confirmed coronavirus infection of different age groups revealed a decrease in the number of germ cells, and their detachment, desquamation, and conglomeration in the lumen of the seminiferous tubules in the complete absence of germ cells. Sertoli cells were absent only in some men of senile age and accompanied by a deterioration in the morphological picture of viral orchitis. In most micropreparations, thickening and loosening of the basement membrane of the seminiferous tubules, Leydig cell hyperplasia, and thickening and edema of the interstitial tissue were noted. In the intertubular space, there are signs of orchitis of predominantly viral etiology: pronounced plasmacytic-lymphocytic infiltration, single neutrophils, endothelitis, and abundant intravascular thrombosis, as well as plethora of hemocapillaries and metachromasia of their inner membrane. Lymphatic vessels are dilated, and there is a large amount of protein in their lumens (Figure 1).

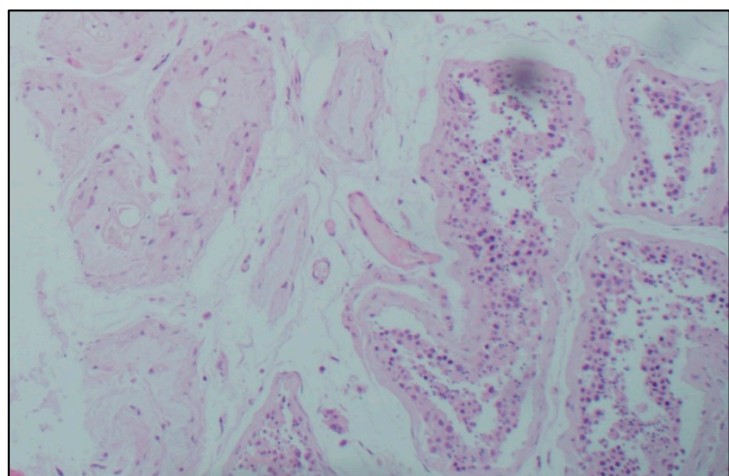

**Figure 1.** Morphological picture of a testicle fragment from a man with a confirmed novel coronavirus infection. Stained with hematoxylin and eosin, magn. $\times 100$.

Immunohistochemical study. IHC-assessment of the testicles of Group I compared with the norm revealed a significant increase in the number of immunopositive germ cells, mainly located in the adluminal compartment of the convoluted seminiferous tubules, stained with antibodies to the SARS-CoV-2 spike protein and nucleocapsid, which was more pronounced in the group of elderly and senile age (Table 2, Figure 2). A similar IHC pattern was observed in Leydig cells and endotheliocytes of peritubular blood vessels (Table 2, Figure 2).

**Table 2.** Number of IHC-positive germ cells, $p < 0.05$.

| Age | *n* | S-Protein (%) | Nucleocapsid (%) | ACE2 (%) |
|---|---|---|---|---|
| control | 20 | – | – | $2.7 \pm 0.1$ |
| 25–44 | 9 | $26.2 \pm 1.2$ [a] | $17.1 \pm 0.8$ [a] | $9.4 \pm 0.4$ [a] |
| 45–60 | 17 | $51.5 \pm 2.5$ [a] | $26.4 \pm 1.3$ [a] | $13.2 \pm 0.6$ [a] |
| 61–75 | 43 | $57.9 \pm 2.6$ [a] | $37.8 \pm 1.6$ [a] | $18.3 \pm 0.9$ [a] |
| 76–90 | 21 | $65.4 \pm 3.2$ [a] | $59.5 \pm 2.9$ [a] | $25.7 \pm 1.1$ [a] |

[a] experimental groups compared with control, $p < 0.05$.

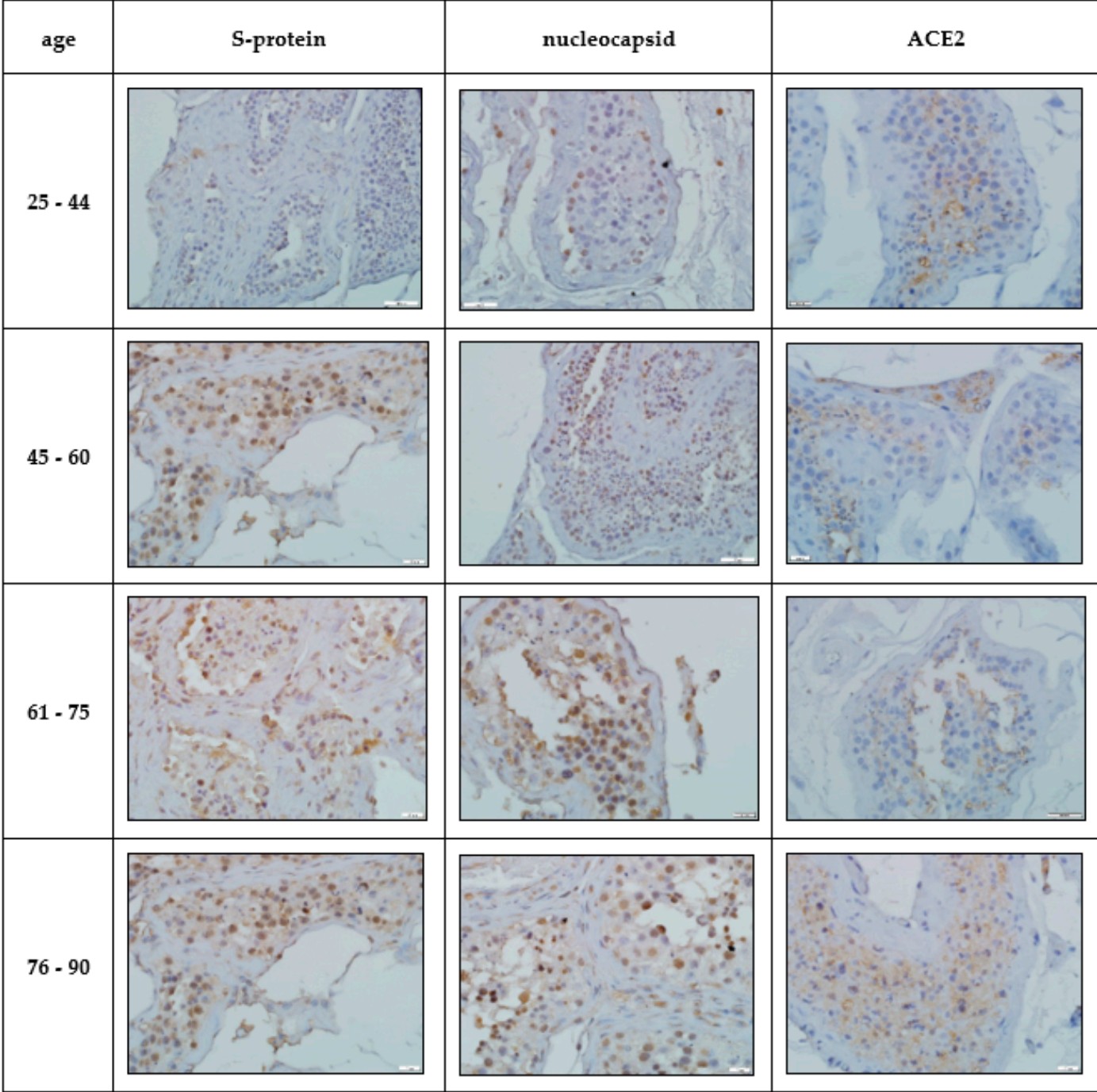

**Figure 2.** Immunohistochemical pattern of germinal epithelium in patients with confirmed novel coronavirus infection. Stained with hematoxylin, magn. ×200.

In the group of COVID-19-positive patients, a significant increase in the number of IHC-stained ACE2 cells of the late stages of spermatogenesis was revealed compared to the control group, and the largest number of positive cells was detected in the cohort of patients of senile and elderly age (Table 2, Figure 2).

FISH study using confocal microscopy revealed immunolabeling for SARS-CoV-2 in the cytoplasm of spermatogenic cells located mainly in the adluminal compartment of the seminiferous tubules (Figure 3).

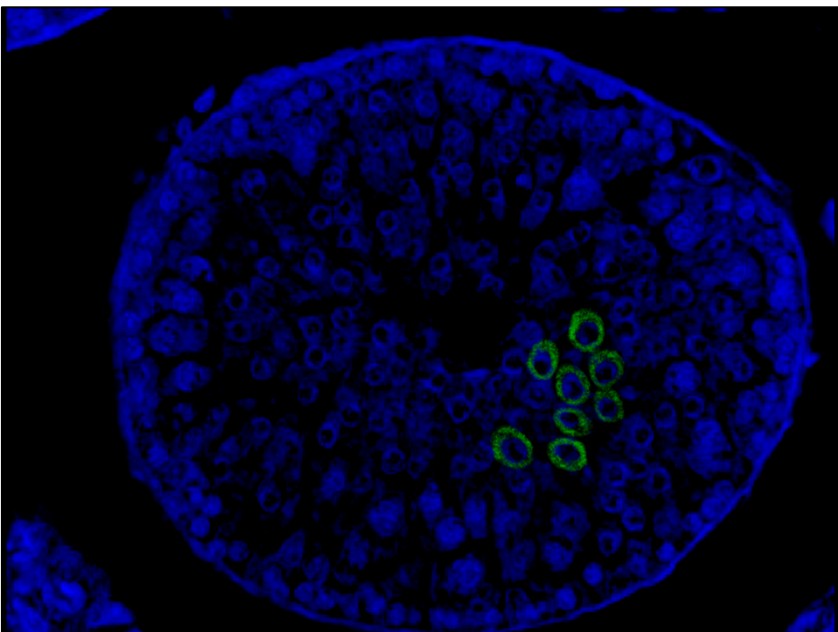

**Figure 3.** Patient P.; 51 years old, FISH study by confocal microscopy. SARS-CoV-2 invasion into germ cells. Visualization with secondary antibodies conjugated with Alexa Fluor 488 (green glow), DAPI core staining (blue glow). Magn. ×200.

## 4. Discussion

The present study is devoted to the immunohistochemical evaluation of impaired spermatogenesis after SARS-CoV-2 invasion into the testicles of men of different age groups.

The violation of the histoarchitectonics of the seminiferous tubules and signs of hypospermatogenesis found in all age groups indicate a decrease in the functional status in patients with COVID-19 infection. This occurs as a result of damage to the wall of blood vessels and inflammatory infiltration (viral orchitis). This leads to a decrease in the trophic function of the blood-testicular barrier due to thrombosis of blood vessels and ischemia [13]. The predominance of lymphoid immunocompetent cells in micropreparations also confirms the viral origin of the observed inflammatory changes [14]. Interestingly, the decrease in the number or absence of Sertoli cells in single senile men in Group I was accompanied by a deterioration in the morphological picture of viral orchitis. Given the role of Sertoli cells in the formation of the blood–testis barrier, a decrease in their number leads to a violation of the germinal function.

An IHC study of testicular autopsies from COVID-19-positive patients revealed an increase in the number of S-protein- and nucleocapsid-positively stained germ cells, which indicates SARS-CoV-2 invasion into them. At the same time, the number of positive cells was significantly higher in elderly and senile people. This is due both to a gradual physiological decrease in the ability to reproduce and to a deterioration in the immune status of patients in this group [15,16].

It is known that the ACE2 receptor is one of the key ones through which SARS-CoV-2 penetrates into the host cell [17]. In our study, an increase in the number of ACE2-positive germ cells was accompanied by a deterioration in the degree of hypospermatogenesis. In the group of patients with confirmed coronavirus infection older than 45 years, the decrease in spermatogenic function was more pronounced than in the cohort of young people, which is also associated with the peculiarities of physiological spermatogenesis [16,18].

In our study, Leydig cells immunostaining was found, the main function of which is the synthesis of testosterone. Thus, we can talk about the presence of direct and indirect mechanisms leading to disruption of spermatogenesis. The direct effect is associated with the invasion of SARS-CoV-2 into germ cells, and the indirect effect is a damaging effect on interstitial endocrinocytes. This leads to germinal and endocrine dysfunction.

The subject of further research should be the disclosure of the leading mechanisms of SARS-CoV-2 interaction with ACE2 receptors and its effect on the reproductive function of patients with COVID-19 infection.

Limitations. Despite the fact that a probable correlation between ACE and the degree of hypospermatogenesis was found, and also given the physiologically reduced spermatogenesis in elderly patients, this heterogeneous situation did not allow for a full-fledged objective statistical analysis.

**5. Conclusions**

The results of this study indicate that the viral load of SARS-CoV-2 on the testicles, estimated by the number of S-protein-, nucleocapsid-, and ACE2-positive germ cells, is more pronounced in the group of elderly patients.

**Author Contributions:** Conceptualization—G.A.D. and D.B. Methodology, Investigation, Resources—G.A.D., D.B. and V.S. Project administration—P.S. and A.K. Data curation, Formal analysis—D.B. and M.V. Writing—original draft, Visualization—M.V. and D.B. Writing—review & editing—G.A.D., E.K. and T.D. All authors have read and agreed to the published version of the manuscript.

**Funding:** This research received no external funding.

**Institutional Review Board Statement:** The study was conducted in accordance with the Declaration of Helsinki, and approved by the Institutional Ethics Committee of Sechenov University (protocol No. 162 of 11 February 2022).

**Informed Consent Statement:** Informed consent was obtained from relatives or official representatives of all subjects involved in the study.

**Data Availability Statement:** All data and materials, as well as software application or custom code, support their published claims, comply with field standards, and are openly available.

**Conflicts of Interest:** The authors declare no conflict of interest.

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
