# Peer review of "Immunohistochemical Analysis of Spermatogenesis in Patients with SARS-CoV-2 Invasion in Different Age Groups"

_cimb, doi:10.3390/cimb45030159_

Round 1

Reviewer 1 Report

Useful clinical information but a shortfall exists within the manuscript

Small numbers is a limitation and other limitations such as heterogeneity need to be disclosed before the conclusion

Other limitation includes lifestyle smoking medical histories unavailable. Did the elderly die purely of covid infection or had multiple medical problems? If so this should be stated as a limitation.

An increase in the level of ACE2 expression directly correlates with the degree of impaired spermatogenesis. This fact is not well established apart from covid study observations and needs to be reviewed with more robust references, or shaded.

Physiological decrease in the ability to reproduce, and to a deterioration in the immune status of patients in this group. Needs referencing

Sertoli cells were absent only in some men of senile age- how did this affect the extent of covid damage? Was it significant in elderly men?

In the group of COVID-19-positive patients, a significant increase in the number of IHC-stained for ACE2 cells of the late stages of spermatogenesis was revealed compared to the control group, and the largest number of positive cells was detected in the cohort of patients of senile and elderly age (Table 2, Fig. 2). It seems this is more subjective and needs clarification on how counting ACE2 cells in a heterogeneous situation can be made uniform throughout for proper statistical analyses.

Show statistical outcomes in Table 2. It simply states the Number of IHC-positive germ cells, p<0.05. It is unclear which parts are significant to what?

IHC-assessment of the testicles of group I compared with the norm revealed a significant increase in the number of immunopositive germ cells, Is this shown in tabular form?

They could do with Table 3 showing the counting of cells between the 2 age demarcations along with the statistical significance

Significant improvements needed

Reviewer 2 Report

In this manuscript, the authors have reported results demonstrating evaluate immunohistochemical changes in spermatogenesis during SARS-CoV-2 in-15 vasion in different age groups. The study can be greatly improved if the following suggestions were incorporated.

1-           Please add some studies related to GENE ANALYSIS OF SPERMATOGENESIS IN PATIENTS in the main introduction.

2-           Line 72, sentence “Causes of death in men of elderly…” and line 198 “Of particular note is the presence of the IHC reaction in Leydig…” are not meaningful. Please check the grammar.

3-           Please provide the staining method and add the details of probe information in the methods section.

4-           Line 181, the sentence “This leads to a decrease in the trophic...” is not cited.

Round 2

Reviewer 1 Report

no further comments

Reviewer 2 Report

All parts that the reviews mentioned have been improved. I do not have any future comments.